# Values and Preferences Related to Cancer Risk among Red and Processed Meat Eaters: A Pilot Cross-Sectional Study with Semi-Structured Interviews

**DOI:** 10.3390/foods10092182

**Published:** 2021-09-14

**Authors:** Victoria Howatt, Anna Prokop-Dorner, Claudia Valli, Joanna Zajac, Malgorzata M. Bala, Pablo Alonso-Coello, Gordon H. Guyatt, Bradley C. Johnston

**Affiliations:** 1Department of Community Health and Epidemiology, Dalhousie University, Halifax, NS B3H 4R2, Canada; vhowatt@dal.ca; 2Chair of Epidemiology and Preventive Medicine, Department of Medical Sociology, Jagiellonian University Medical College, 31-008 Krakow, Poland; anna.prokop@uj.edu.pl; 3Department of Paediatrics, Obstetrics, Gynaecology and Preventive Medicine, Universidad Autónoma de Barcelona, 08193 Bellaterra, Spain; claudia.valli89@gmail.com; 4Iberoamerican Cochrane Centre Barcelona, Department of Clinical Epidemiology and Public Health, Biomedical Research Institute Sant Pau (IIB Sant Pau), 08041 Barcelona, Spain; palonso@santpau.cat; 5Department of Hygiene and Dietetics, Epidemiology and Preventive Medicine, Jagiellonian University Medical College, 31-008 Krakow, Poland; joanna.faustyna.zajac@gmail.com; 6Chair of Epidemiology and Preventive Medicine, Department of Hygiene and Dietetics, Jagiellonian University Medical College, 31-008 Krakow, Poland; malgorzata.1.bala@uj.edu.pl; 7CIBER de Epidemiología y Salud Pública (CIBERESP), 08023 Barcelona, Spain; 8Department of Health Research Methods, Evidence and Impact, McMaster University, Hamilton, ON L8S 4K1, Canada; guyatt@mcmaster.ca; 9Department of Medicine, McMaster University, Hamilton, ON L8S 4K1, Canada; 10Departments of Nutrition, Epidemiology and Biostatistics, Texas A&M University, College Station, TX 77843, USA

**Keywords:** red meat, processed meat, values, preferences, cancer risk, survey

## Abstract

**Introduction:** Over the last decade, the possible impact of meat intake on overall cancer incidence and mortality has received considerable attention, and authorities have recommended decreasing consumption; however, the benefits of reducing meat consumption are small and uncertain. As such, individual decisions to reduce consumption are value- and preference-sensitive. Consequently, we undertook a pilot cross-sectional study to explore people’s values and preferences towards meat consumption in the face of cancer risk. **Methods and analysis:** The mixed-method pilot study included a quantitative questionnaire followed by qualitative evaluation to explore the dietary habits of 32 meat eaters, their reasons for eating meat, and willingness to change their meat consumption when faced with a potential risk reduction of cancer over a lifetime based on a systematic review and dose–response meta-analysis. We recruited a convenience sample of participants from two Canadian provinces: Nova Scotia and Prince Edward Island. This project was approved by the Research Ethics Board for Health Sciences research at Dalhousie University, Canada. **Results:** The average weekly consumption of red meat was 3.4 servings and the average weekly consumption of processed meat was 3 servings. The determinants that influenced meat intake were similar for both red and processed meat. Taste, cost, and family preferences were the three most commonly cited factors impacting red meat intake. Taste, cost, and (lack of) cooking time were the three most commonly cited factors impacting processed meat intake. None of the participants were willing to eliminate red or processed meat from their diet. About half of participants were willing to potentially reduce their meat consumption, with one third definitely willing to reduce their consumption. **Strengths and limitations:** This study is the first that we are aware of to share data with participants on the association of red meat and processed meat consumption and the risk of cancer mortality and cancer incidence, including the certainty of evidence for the risk reduction. The limitations of this study include its small sample size and its limited geographic sampling. **Conclusions:** When presented explicit information about the small uncertain cancer risk associated with red and processed meat consumption, study participants were unwilling to eliminate meat, while about one-third were willing to reduce their meat intake.

## 1. Introduction

Nutrition guidelines, including Canada’s Food Guide, provide important directions for community and institutional food programs and are important tools for the promotion of healthy eating [1]. Nutritional choices are important for general health and may help prevent major illnesses, including cancer [2]. Recently, the possible influence of unprocessed red and processed meat on overall cancer incidence and mortality has received attention [3,4,5]. Unprocessed red meat (hereafter, referred to as red meat) is typically defined as any type of meat from mammals (e.g., beef, pork, lamb), whereas processed meats are defined as red or white meats preserved by smoking, curing, salting, or by the addition of preservatives [6].

Members of the public are key stakeholders for nutrition guidelines because they are ultimately left with the choice of whether or not to adhere to their recommendations [7,8]; however, to our knowledge, aside from the recent NutriRECS guideline on red and processed meat [9], all nutrition guidelines have been developed without explicit consideration or the systematic assessment of public values and preferences [7,8,10,11,12,13]. For example, the process of developing the recent Canadian Food Guidelines included public consultation about the proposed guidelines [14]; however, it is unclear how public comments and input was received, prioritized, and integrated. In addition, most guidelines suffer from a lack of systematic engagement and integration of values and preferences from members of the public [13].

The decision to modify dietary intake may be associated with despondency or pleasure that ultimately affects an individual’s overall satisfaction and quality of life [15]. Indeed, people generally have considerable reluctance or difficulty in changing either the amount or type of food they consume [15]. The values of community members with respect to nutritional issues are uncertain. For example, it is not clear how high the risk of major adverse future events, such as cancer, would have to be to motivate people to change their diet. A recent systematic review on values and preferences regarding consumption of meats of different types and health risks found that, based on a low certainty of evidence, people are attached to their general meat consumption and are typically unwilling to change their consumption for health reasons [16]. We wanted to assess this by directly asking omnivores to consider the factors (e.g., cost and taste) impacting their red and processed meat consumption and weigh them against cancer risk reduction based on up-to-date systematic summary data tailored to their typical weekly consumption of red or processed meat.

## 2. Methods

### 2.1. Design

This study is a pilot study featuring mixed methods.

### 2.2. Protocol Development

Based on the results of a rigorous systematic review and meta-analysis on the risk of cancer associated with red and processed meat intake [17], we developed a survey and semi-structured interview guide to elicit the values and preferences of participants regarding red and processed meat intake [18]. The study included a quantitative assessment using a questionnaire and direct choice exercise. This was followed by qualitative evaluation through semi-structured interviews to explore participants’ reasons for eating meat, as well as factors influencing their willingness to reduce or stop eating meat. The Research Ethics Board for Health Sciences research at Dalhousie University approved this project.

### 2.3. Participant Selection

Eligible participants were those who lived in Atlantic Canada (Nova Scotia, New Brunswick, Prince Edward Island, and Newfoundland and Labrador) and consumed red or processed meat. We recruited a convenience sample of participants using social media posts and community posters in the Halifax Regional Municipality. In addition, to help recruit those residing in Nova Scotia, flyers were distributed to local participants in the Canadian Longitudinal Study for Aging [19]. Participants were included if they consumed at least 1 weekly serving of either red meat or processed meat and were between the ages of 18 and 80. Participants were excluded if they were pregnant, had suffered a major cardiovascular event, or had ever been diagnosed with cancer.

### 2.4. Procedures

#### 2.4.1. Direct-Choice Exercise

Participants completed a questionnaire that included a direct-choice exercise. First, we collected basic demographic and medical history information (e.g., age, sex, family history of cancer, current consumption of red meat, processed meat, and typical source of red and processed meat). Participants then completed a dietary survey describing their current meat intake, whether they take health risks into account when choosing their diet, and whether their food choices affect other people (e.g., preparing food for children or others in the household). We assured participants that there were no correct answers, and that we wanted to hear their genuine opinions. To standardize questions and elicit consumption patterns, we showed participants pictures illustrating types of meats and serving sizes (Appendix A).

Subsequently, using standard serving sizes of 120 g for red meat and 50 g for processed meat, we showed participants data illustrations generated using MAGICapp (https://app.magicapp.org/app#/guidelines, Accessed on 22 August 2021). These graphics were based on our meta-analytical data to illustrate the effect of eliminating the consumption of red and processed meat on the risk of dying from or developing cancer compared to average consumption [17]. The MAGICapp graphics also contained the certainty of the evidence using the GRADE approach to help inform participants’ understanding of the quality of evidence we presented to them (Appendix A). We tailored the risk reduction based on the reported weekly intake for each participant. For instance, if the participant consumed four servings per week of red meat, we presented the risk reduction associated with a reduction of four servings.

We then elicited the participant of willingness to: (i) eliminate meat, and, if they were unwilling to eliminate, to (ii) reduce their meat consumption when faced with an absolute risk reduction of overall (all-cause) cancer mortality and the lifetime risk of a diagnosis of overall cancer (incidence). Participants then ranked their level of willingness to eliminate or reduce their intake on a 1–7 Likert scale, with 1 meaning “definitely not willing to eliminate (or reduce)”, 2 or 3 meaning “somewhat not willing to eliminate (or reduce)”, 4 or 5 meaning “somewhat willing to eliminate (or reduce)”, and 6 or 7 meaning “definitely willing to eliminate (or reduce)”.

#### 2.4.2. Semi-Structured Interview

A semi-structured interview was completed either in-person or via private video conferencing. We discussed the factors that influenced participants red and processed meat intake using open-ended questions. We also asked them how the MAGICapp figures presented to them impacted their decisions, if at all. We then discussed what the determinants were for their decision regarding changing or not changing their meat consumption patterns by referring to the Likert scale from the direct-choice exercise.

### 2.5. Data Synthesis and Analysis

#### 2.5.1. Quantitative Analysis

We calculated means and standard deviations for participant demographic information, the number of servings of red and processed meat they consumed, and their willingness to eliminate or reduce their consumption.

#### 2.5.2. Qualitative Analysis

A research assistant (VH) recorded audio, transcribed the semi-structured interviews in verbatim and used thematic analysis for the qualitative analysis [20,21]. We developed a codebook based on the participant answers. With the help of an experienced qualitative researcher (APD), the research assistant then coded text segments of the interview transcripts to represent units of meaning using MaxQDA 2018 software package (maxqda.com, Accessed: Aug 22, 2021). Next, we analyzed code reports, displayed, summarized and compared coded excerpts between interviewees, and wrote memos to track observed regularities and dissimilarities. Each of these steps enabled us to identify emerging patterns and themes. The themes were then explored and contextualized with individual characteristics of the interviewees, such as age, sex, and willingness vs. unwillingness to eliminate or reduce red or processed meat consumption.

## 3. Results

### 3.1. Participant Demographics

We recruited and interviewed 32 participants living in Nova Scotia and Prince Edward Island, Canada, in the summer of 2019. They ranged from 21–79 years old, with the largest proportion of individuals in the 23–27 age range (n = 10, 31%, Table 1). Of these, 18 were male and 14 were female. All participants completed the survey and semi-structured interview. The average weekly consumption of red meat was 3.4 servings (SD 1.7 to 5.1), and the average weekly consumption of processed meat was 3 servings (SD 0.9 to 5.1).

Of the 32 participants, 31 consumed at least 1 weekly serving of red meat, and 15 consumed at least 1 weekly serving of processed meat. Fourteen of the 15 participants who consumed at least 1 weekly serving of processed meat also consumed at least 1 weekly serving of red meat.

### 3.2. Willingness to Eliminate or Reduce Red Meat

Twenty-eight (90%) of 31 participants were not willing to eliminate their red meat consumption (between 1 and 3 on the Likert scale). Three (9.7%) participants were “somewhat willing” to eliminate (4 or 5 on the Likert scale) while no participants were “definitely willing” to eliminate their red meat consumption (Figure 1).

In contrast, when asked about the reduction of meat intake, 15 (48.4%) of 31 participants were willing to potentially reduce their red meat consumption. Six (19%) were “definitely willing” to reduce (6 or 7 on the Likert scale), nine (29%) participants were “somewhat willing” to reduce eating red meat (4 or 5 on the Likert scale). The remaining 17 (46%) were “somewhat unwilling” to “definitely unwilling” to reduce their red meat consumption (Figure 1). An approximate equal number of these participants were male and female. Half were also between 18 and 40 years old, with one participant between 41 and 60 years old and the remaining five were in the 61–80 year old category. Three were employed for wages, while four were students and the remaining five were retired or not working.

### 3.3. Willingness to Eliminate or Reduce Processed Meat

In total, 13 (86.7%) of the 15 participants who ate processed meat were not willing to eliminate their meat consumption (between 1 and 3 on the Likert scale). Two (13.3%) were “somewhat willing” to eliminate (4 or 5 on the Likert scale) and no participants were “definitely willing” to eliminate their processed meat consumption (Figure 2).

By contrast, when asked about reduction of meat intake, eight (53.3%) of 15 participants were potentially willing to reduce their processed meat consumption. Four (26.7%) of 15 participants were “definitely willing” to reduce their intake while the remaining four (26.7%) participants were “somewhat willing” to reduce eating processed meat (Figure 2). The age groups were approximately evenly distributed, with two between 18 and 40, one between 41 and 60 and two between 61 and 80. Similarly to red meat, one of these participants was employed, two were students, and the remaining three participants were either retired or not currently working.

### 3.4. Preferences and Factors Impacting Meat Consumption in the Face of Cancer Risk Reduction

The determinants that influenced participant meat intake were similar for both red and processed meat. Taste, cost, and family preferences were the three most commonly cited factors impacting red meat intake. Similarly, taste, cost, and cooking time were the three most commonly cited factors impacting processed meat intake. While cost was cited as a top factor for both, people tended to purchase less red meat and more processed meat because of the difference in expense. Similarly, health was seen as a positive factor for consuming red meat for the iron and protein content, whereas, for processed meat, it was seen as a negative factor due to the addition of preservatives, and often high sodium content. Some quotations from the participants illustrate the pattern of preferences related to meat consumption (Table 2).

Based on the semi-structured interviews, participants outlined several key reasons for consuming red and processed meat. Table 3 provides quotations supporting our findings. Participants chose red and processed meats mainly for their unique flavors, which “gives satisfaction” and “cannot be replaced” by any other types of meat or alternatives. Many of the interviewees’ inclinations for meat were related to eating habits from their family homes, to which interviewees continued to conform as adults. Preferences of nuclear family members were also influential. Partner preferences towards meat were negotiated and particular strategies of managing meat consumption were developed in some interviewees’ households. Participants also suggested that the broader social context can facilitate eating red and processed meat. A social practice of preparing and consuming meat during particular social events, i.e., barbecue parties, was a key pattern and participants often felt obliged to eat customary meat dishes.

Some participants talked about price as a barrier to eating red meat. On the contrary, the lower cost of processed meat encouraged more regular consumption for some participants. Another pragmatic reason to consume processed meat was convenient access and simple preparation. On a busy day, “easy grab and go snacks”, including processed meat was, for some participants, optimal for satisfying hunger. When talking about reasons to consume red meat, some mentioned health factors, including meat as part of a balanced diet with substantive nutritional value. Those responses corresponded with recalling claims about red meat being a part of a healthy diet. Finally, some participants perceived negative environmental consequence of meat production as a motivation to reduce eating meat.

Study participants who were willing to reduce their red or processed meat intake cited several reasons for their decision. Some viewed the evidence presented to them as convincing and “something to be concerned about.” They also referred to common knowledge regarding processed foods as being generally unhealthy as a reason to reduce their intake of processed meat. Others used the evidence shown to them as further motivation to change their consumption patterns, as they were already considering a possible change before we interviewed them. One participant was especially concerned about the environmental impact of meat consumption and reported that the evidence presented could “help push [them] over the edge” to reduce their consumption.

## 4. Discussion

### 4.1. Main Findings

The 32 participants in Atlantic Canada who informed us on their willingness to eliminate or reduce their meat intake based on the best available systematic review evidence, consumed on average 3.4 servings of red meat and 3 servings of processed meat per week. Based on our results, the main determinants of dietary behavior that influenced red and processed meat intake, included taste, cost, cultural habits, and ease access or preparation time. About half of the participants were willing to potentially reduce their intake, with one third of participants stating that they were “definitely willing” to reduce eating meat; however, none of the participants were willing to completely stop their meat consumption. Although we had a limited sample size, results were generally consistent regardless of age, sex, and employment status (e.g., student, worker, retired) with respect to the amount of meat consumed weekly and participant willingness to eliminate or reduce meat intake.

Many participants who were willing to reduce their red or processed meat intake were already conscious of current health trends with respect to meat consumption based on mainstream media reports, with particular attention paid to perceived increased cancer risk associated with higher levels of consumption. Several participants mentioned that they would be more willing to reduce their processed meat intake because of this increased attention [3,22]; however, most participants valued the taste of meat, their family preferences, or the low cost of meat more than evidence demonstrating a possible risk reduction in cancer, suggesting an overall unwillingness to consider making a change. Many of these participants were “definitely unwilling” to change their consumption patterns and several could not think of any reasons that would lead them to consider making a change in the future.

### 4.2. Comparison to Other Similar Studies

A recent systematic review on values and preferences regarding consumption of meats of different types and health risks found that, based on a low certainty of evidence, people are highly attached to their general meat consumption and are unwilling to change their consumption for health reasons [16]. We wanted to assess this by directly asking omnivores to consider factors (e.g., cost, taste) impacting their red and processed meat consumption and weigh them in terms of cancer risk reduction based on an up-to-date systematic summary of data tailored to their typical weekly consumption. While dietary guidelines often provide opportunity for public feedback on the initial guideline structure and objectives, as well as the preliminary findings [2,5], to our knowledge they do not incorporate public values and preferences based on the estimated risks and corresponding certainty of evidence. Based on guidance from the GRADE working group and the National Guideline Clearinghouse Extent Adherence to Trustworthy Standards (NEATS) on optimal guideline methods, our study design and findings on value and preferences should be of interest to those making dietary recommendations on red and processed meat [23,24].

### 4.3. Strengths and Limitations

Our study has a number of strengths. First, we shared real data from a high-quality systematic review of red meat and processed meat consumption and risk of cancer mortality and cancer incidence studies [9,17]. Second, we presented the certainty of evidence for the association of meat intake and cancer risk generated using the GRADE approach. Based on previous research, we anticipated that if participants were shown the best estimate of cancer reduction, as well as the certainty of this reduction, which was low to very low based on the GRADE assessments, their responses may be different than if they were shown only the risk reduction. Third, using the dose–response meta-analysis data from a systematic review [17], we tailored the reduction in cancer risk over a lifetime for each participant based on their reported number of servings of red and processed meat per week. Many participants stated that the evidence was not conclusive enough to warrant a change in meat consumption habits, suggesting that an understanding of the limitations of the evidence on meat intake and cancer risk, primarily based on observational data, which is at risk of confounding and thus lower certainty, is informative when making nutrition decisions.

The limitations of this study include its sample size of only 32 participants, its limited geographical representation, its uneven age distribution, and its focus on the impact of meat on cancer risk with omission of cardiovascular risk. Regarding the omission, although we conducted a systematic review and dose–response meta-analysis on cardiovascular disease [25], the evidence was less compelling when compared to cancer risk [17]. This was a pilot study to test our study methods and to develop our study protocol for a larger study [18]. An additional limitation of note is that after presenting the potential risk reductions together with the uncertainty of evidence, rather than asking participants about their willingness to both decrease and increase their meat intake, we only asked participants if they were willing to decrease their meat intake. In the qualitative interviews, although we did not directly ask participants, none mentioned that they may be inclined to increase their intake after seeing the uncertainty of evidence.

### 4.4. Implications

Although we are the first to collect value and preference data based on a systematic review of the evidence on red meat and processed meat consumption and potential cancer risk, the generalizability of our results is limited. To help overcome this limitation and explore the consistency of our findings, we aim to also conduct this study in multiple countries [18]. To improve adherence and optimally develop dietary guidelines, as is done in evidence-based clinical practice, patient and public values and preferences should be closely considered. Ideally, dietary guidelines should be based on the best summaries of the absolute risks and the corresponding certainty of the evidence for the potential risks associated with meat consumption. Indeed, participants demonstrated that they are interested in the level of certainty, and that certainty of evidence played a role in their decision-making process.

## 5. Conclusions

When presented explicit information about cancer risk reduction associated with red and processed meats, omnivores valued competing factors such as the taste, cost, family and cultural norms, and ease of preparation over the small uncertain risk of cancer. Study participants were unwilling to eliminate meat, while about one-third were definitely willing to reduce their meat intake. Although we had a limited sample size, the results were generally consistent regardless of age, sex, and employment status (student, employed, and retired) with respect to the amount of meat consumed weekly and participant willingness to eliminate or reduce their meat intake.

## Figures and Tables

**Figure 1 foods-10-02182-f001:**
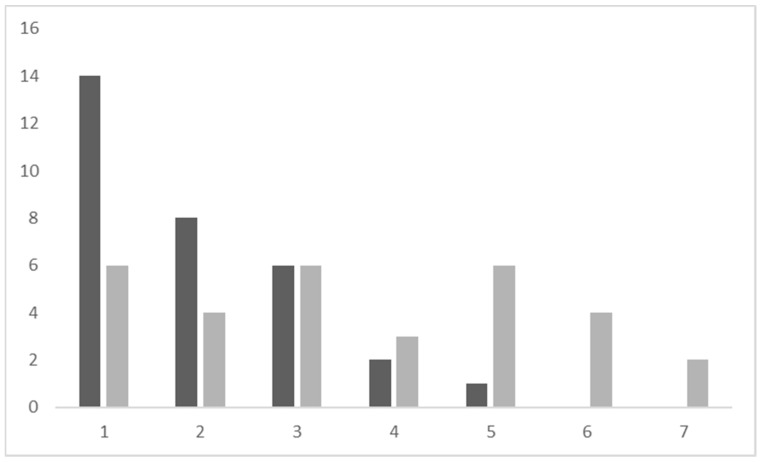
Participant willingness to eliminate (**dark bars**) or reduce (**light bars**) red meat according to ranking on a Likert scale (1–7). The vertical axis refers to a number of participants. The horizontal axis refers to the Likert scale.

**Figure 2 foods-10-02182-f002:**
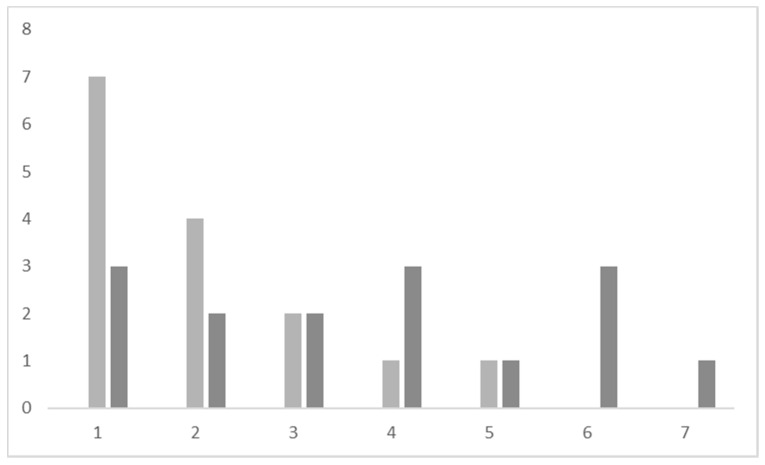
Participant willingness to eliminate (**light bars**) or reduce (**dark bars**) processed meat according to ranking on a Likert scale (1–7). The vertical axis refers to the number of participants. The horizontal axis refers to Likert scale.

**Table 1 foods-10-02182-t001:** Demographic characteristics of participants.

Study Participant No.	Age	Sex	Ethnicity	Level of Education	Employment Status	Marital Status	Family Characteristics	Exercise/Sport
1	24	M	European origins	Master’s	Student	Married	No children	Regularly
2	36	M	European origins	Master’s	Student	Married	1 child	With some regularity
3	23	F	European origins	Bachelor’s	Student	Single	No children	With some regularity
4	23	M	European origins	Bachelor’s	Student	Single	No children	Regularly
5	23	M	European origins	Bachelor’s	Student	Single	No children	Regularly
6	30	F	European origins	Bachelor’s	Student	Living common law	No children	Regularly
7	23	M	European origins	Bachelor’s	Student	Single	No children	Regularly
8	27	F	European origins	Master’s	Student	Living common law	No children	Regularly
9	23	F	Middle Eastern origins	Bachelor’s	Student	Single	No children	Never
10	29	M	European origins	Master’s	Student	Married	No children	Regularly
11	32	F	Middle Eastern origins	MD	Student	Married	2 children	Seldom
12	25	M	European origins	Bachelor’s	Student	Single	No children	With some regularity
13	48	F	European origins	Bachelor’s	Employed for wages	Married	1 child	Never
14	49	M	European origins	Bachelor’s	Employed for wages	Married	1 child	Regularly
15	47	M	African origins	Master’s	Employed for wages	Single	No children	Seldom
16	23	F	European origins	Master’s	Employed for wages	Single	No children	Seldom
17	51	M	Middle Eastern origins	Bachelor’s	Employed for wages	Living common law	3 or more children	With some regularity
18	76	M	Other North American	Secondary education	Retired	Married	No children	Regularly
19	79	F	European origins	Secondary education	Retired	Married	No children	Regularly
20	72	M	European origins/African origins	Secondary education	Retired	Married	No children	Seldom
21	71	F	European origins	Secondary education	Retired	Married	No children	regularly
22	49	M	European origins	Secondary education	Employed for wages	Married	2 children	seldom
23	33	F	European origins	Secondary education	Employed for wages	Married	2 children	seldom
24	67	F	European origins	Bachelor’s	Retired	Divorced	No children	Regularly
25	23	F	European origins	Bachelor’s	Employed for wages	Living common law	No children	With some regularity
26	79	M	European origins	Secondary education	Retired	Married	No children	regularly
27	72	F	European origins	Secondary education	Homemaker	Married	No children	With some regularity
28	35	M	Asian origins	Master’s	Employed for wages	Married	1 child	Regularly
29	21	F	European origins	Bachelor’s	Out of work and looking for work	Single	No children	With some regularity
30	27	M	European origins	Master’s	Employed for wages	Single	No children	Regularly
31	36	M	European origins	Master’s	Employed for wages	Married	1 child	With some regularity
32	30	M	European origins	Bachelor’s	Employed for wages	Single	No children	Regularly

**Table 2 foods-10-02182-t002:** Factors that impacted unprocessed red and processed meat intake.

	Unprocessed Red Meat		Processed Meat	
Factors	Number of Mentions	Example Quotations	Number of Mentions	Example Quotations
Taste	24	“Taste because it tastes good. If I had to compare that with other forms of protein like chicken and stuff like that, well cooked red meat really tastes better which gives me satisfaction from eating.” (M, Age 35, Employed)	10	“Especially charcuterie and salamis, I enjoy those a lot.” (F, Age 27, Student)
		“As a baseline I find that red meat tastes better than white meat in almost any circumstance.” (M, Age 29, Student)		“I think the both of us (referring to wife) like ham and we prefer that one over other types of processed meats. It is fairly lean to us.” (M, Age 76, Retired)
Cost	17	“...that might be the most limiting factor for why I don’t eat maybe another serving of red meat a week. It is generally more expensive than both processed meat and poultry and fish.” (M, Age 27, Employed)	8	“I’m eating a lot of sandwich meats to make lunches for school being a student trying to find cheap meats for lunches.” (M, Age 23, Student)
		“Because of a limited income on a pension, the [high] cost is a big factor.” (M, Age 76, Retired)		“I tend to buy them only because they’re cheap and because they’re quick to prepare, so I purchase them when time is short and money is tight.” (F, Age 48, Employed)
Health	12	“As far as red meat, with health, I have a really difficult time getting things like iron and protein and while I am aware that you can get these things in other foods like eggs and vegetables, between weird allergies and personal preference, it is easier to get these things out of meat products.” (F, Age 21, Unemployed)	2	“I’d be looking at preservatives knowing they’re not great for us in large quantities. We don’t need all that extra stuff.” (F, Age 23, Student)
		“In terms of health, I consider red meat an essential aspect of a balanced diet. It is a good source of iron for the children when it is lean beef, and it is a good source of nutrition.” (F, Age 32, Student)		“Generally, it has been hampered that processed foods are less healthy.” (M, Age 36, Student/Employed)
Family Preferences/Tradition	20	“For tradition, it is more so that it is how my parents have eaten for a long time. I live at home and abide by those rules and none of us have been into veganism. I am not the one paying for the food, while I help, from the standpoint of what’s in the fridge, I’m not going to complain and there is no moral shift from what my parents eat.” (M, Age 23, Student)	5	“I would personally not buy ham, but my mother does, so I eat it because of that.” (F, Age 23, Student)
		“In terms of family preferences, I lived with my dad most of the time when I was younger and red meats would make up most of the meals in a week, so I grew up with it.” (M, Age 29, Student)		“Because of the way it tends to work in the house, my wife comes up with meal plans and I execute.” (Cited same reason for both unprocessed red and processed meat) (M, Age 36, Student/Employed)
Convenience/Availability	6	“It is very easily available in the meat section of any grocery store you go to, so you don’t have to go searching for it, so that kind of influences your buying decisions because it stares you in the face.” (M, Age 35, Employed)	7	“When you go in any store, you will see pork on sale. If it is in Superstore this week, it will be on sale at Walmart the next week. For example, this week Superstore, the advertisement stated that the price was reduced for pork. Next week, it will be Sobey’s.” (M, Age 47, Employed)
		“In terms of availability, it is usually more easily and widely available in the places I go a lot and it is usually cheaper and more widely available and it is just easier to prepare and usually the convenient option and you don’t run into time and funds issues.” (F, Age 21, Unemployed)		“They are prepared, easy to use, readily available. It makes it easy to have those.” (F, Age 27, Student)
Cooking Time	2	“I think that for me, on a practical level, the cooking time is a big factor, like recipes. If I want to eat less meat, I have to work at developing new recipes because most of the things I normally make include meat in them.” (M, Age 25, Student)	9	“It is easier to not cook processed sandwich meats that are ready made.” (M, Age 23, Student)
		“I’m always a hyper and want to get things done person. With cooking time, the faster things get done, the faster I eat [unprocessed red meat is a relatively quick meal to prepare compared to alternatives].” (M, Age 76, Retired)		“My son and I will often go on birding trips, so I can make a lot of sandwiches and it is inexpensive, quick to do, but I have been doing other things to give us more energy like taking boiled eggs. I use pepper not salt.” (M, Age 79, Retired)
Environmental Concerns	3	“I know how bad beef is for the environment. So that makes me eat less.” (F, Age 23, Employed)	1	“Environmental aspect, I have been told on several different occasions that producing processed meat has a large carbon footprint and is contributing to negative environmental issues.” (M, Age 25, Student)
		“The effects that the animal industry has on the planet given the climate crisis right now. We have to start doing something. This would be what pushes me over the edge to stop eating it.” (F, Age 23, Student)		
Social Context	2	“The social context, it makes me eat more because family barbeques usually include steaks or hamburgers.” (M, Age 24, Student)	1	“It is not generally something I pick first. I don’t do it by myself for supper, I’ll do it with my friends or be out for supper (charcuterie board). It is a pairing as opposed to a meal. More likely to have a sausage with my friends at a barbeque.” (M, Age 36, Student/Employed)
		“Barbecues in the summer are good times to have red meat and at parties and social situations.” (F, Age 27, Student)		
Religious Beliefs	3	“I am Muslim, so I do not eat pork, so I always check labels for pork and don’t buy processed meat because it usually has pork in it.” (M, Age 51, Employed)	1	“Similarly, I look on the label to make that the processed meat does not have any pork (Islam).” (F, Age 23, Student)
		“We don’t eat pork for religious reasons.” (F, Age 32, Student)		
Other—Dietary considerations	2	“Because I have to eat around 240g of protein per day for the diet I am trying to follow, the fact that the red meat has one of the highest bioavailabilities of the protein ingested plays an important factor to me.” (M, Age 49, Employed)	1	“It comes down to the fact that I eat keto and the only grab and go snack I can have is cheese and meat. With keto, depending on the processed meat we have the higher fat content.” (F, Age 30, Student/Military)
		“With regards to ketogenic diet, it is considered to be 70-80% fat of my daily calorie consumption and red meat has the highest fat to protein ratio.” (F, Age 30, Student/Military)		
Other—Origin	1	“Buying local—the further it gets away from my home, the less I trust it.” (F, Age 67, Retired)	0	

**Table 3 foods-10-02182-t003:** Participant willingness to change their meat intake given the potential cancer risk and the main determinant(s) of their decision.

Participant #	Main Determinant ^1^	Willingness to Reduce	Example Quotations
1	Factors	Definitely	“I like to know the mechanisms, with the data; if I knew how unprocessed meats at the cellular level could result in cancer, then that would be convincing for me.”
		Yes	
2	Evidence	Somewhat	“I can’t tell you for sure what the threshold would be but needs to be more substantial than that. A direct order from a physician would probably do it.”
		No	
3	Factors	Somewhat	“I think that if it weren’t on the menu in restaurants, I wouldn’t order it. If I hadn’t grown up with it, then it would be easier. Socially, barbeques promote a lot of hotdogs and hamburgers.”
		Yes	
4	Evidence	Somewhat	“[I would need to see] very high evidence showing that meat will significantly put you at risk for cancer death and rare cancer incidences.”
		Yes	
5	Factors	Somewhat	“[I would need to have] methods of obtaining proteins and nutrients from other food sources for free or cheaper. I’m not going to eat meat to get protein when I could get it somewhere else for less expensive and cheaper.”
		Yes	
6	Factors	Definitely	“I through many years of trial and error, have found a way of eating that makes me feel more energetic and healthier. My bloodwork reflects this and my diet has high red and processed meat, so I am unlikely to stop or reduce my intake.”
		No	
7	Factors	Definitely	“Yes, the main thing would be the cancer mortality that I was thinking about, but those factors are what would keep me from not eliminating it. Cost is mainly the one: if I could switch all processed to unprocessed, I would, but the price and convenience outweigh this.”
		Yes	
8	Both	Somewhat	“A clear health risk, if the certainty was higher in the association between cancer and red meat consumption. Another factor is if the price were really high, and it weren’t available.”
		Yes	
9	Factors	Somewhat	“Because the study you showed me is definitely showing something, I would consider reducing based on the data you showed me, but because I really like the taste of meat I do not think I am really willing to cut it out of my diet completely.”
		No	
10	Factors	Somewhat	“I think the only thing that would change my intake would be reactive, so if I had some sort of disease would make a difference… I would have to already be there. I would say that if cost continued to go up, I would consider it because beef is getting more and more expensive, so if it continued, I would switch to cheaper alternatives.”
		Yes	
11	Evidence	Somewhat	“I think that the reason that I decided that I wouldn’t stop eating red meat is that the evidence isn’t convincing—it isn’t that strong. The effect and association isn’t that strong and the evidence isn’t very certain, so it’s both of those together.”
		Yes	
12	Factors	Somewhat	“I think that all those factors that we discussed pushed me toward the side of wanting to not reduce, but if I am honest, I don’t think I would ever stop eating it, but it could push me to wanting to reduce red meat/processed meat consumption.”
		Yes	
13	Both	Somewhat	“What influenced my responses to the graphics was the fact that I didn’t feel the differences were significant enough for me to change the factors that influence my meat. I did not feel that the graphics were enough to override them.”
		Yes	
14	Evidence	Definitely	“What affected me there was the very low probability on each of those studies (referring to mortality and incidence scenarios). Probably also the fact that with every new study, things change.”
		No	
15	Evidence	Somewhat	“Not really, when you showed me the pictures, I saw the risk of getting and dying from cancer, which is why I said I want to reduce. When I said I cannot stop, it is because my health does not show any risks right now with that.”
		Yes	
16	Factors	Somewhat	“I think it’s a bit of routine, so I’m in the routine at home right now eating meat, so when I move and switch up my routine that that will be the big push for me to make that change versus trying to incorporate it into my current lifestyle.”
		Yes	
17	Factors	Definitely	“Yes, I like the taste, so I can’t stop eating it.”
		No	
18	Factors	Definitely	“We don’t know, really, what the processed meat is made of and what the ingredients really are, so we have to take a little bit of time and see what we’re eating. I don’t think we would reduce to the extent of not eating it altogether.”
		Yes	
19	Factors	Somewhat	“Yes, I really like hamburgers, so I don’t want to stop eating it.”
		Yes	
20	Evidence	Definitely	“I understand that if there is no cancer in your genes there and I don’t consume that much meat to start with (not as likely to get cancer), so I didn’t really think of those factors.”
		No	
21	Factors	Somewhat	“It is just hard to figure out what you would eat and how much it would cost if you stopped eating meat. It seems like we spend a lot of money on groceries.”
		No	
22	Evidence	Definitely	“No. There wasn’t enough of a reduction to make me want to stop.”
		No	
23	Factors	Somewhat	“Yes, I was thinking maybe I should change. I am so limited to my food right now as it is, so I don’t want to stop (does not eat many vegetables).”
		Yes	
24	Factors	Definitely	“I’m on such a low side of eating red meat, that I still don’t feel the need. People tend to eat more than they say (not understanding what a serving is). I was automatically thinking of the iron thing.”
		No	
25	Factors	Somewhat	“I think if the risk were substantial, I would stop eating it, but I don’t know what I would call substantial. Because my risk reduction would be around 0.4 and 0.2% right now- it would have to be around 10%.”
		Yes	
26	Evidence	Definitely	“Yeah, I started thinking about cutting back because you showed me the graphics and it looks like something to be concerned about. I know I eat too much processed stuff and not enough vegetables and fruit.”
		Yes	
27	Evidence	Definitely	“Yes, it makes a person think that the data does make a difference. I may work on it based off what I saw, which we have been already, so it’s probably not too much more to work on.”
		Yes	
28	Evidence	Somewhat	“[I would need] at least moderate to high quality evidence that cancer incidence and I don’t really care about mortality because incidence comes with morbidity, that there is a definite reduction in incidence without cutting out red meat completely.”
		Yes	
29	Factors	Definitely	“I am more aware of how “bad” processed meats are for you, but the general convenience associated with it, is something that it more often than not outweighs the consequences of them. The presented risks don’t seem quite enough to outweigh the consequences to eating them. I also seem to take in enough of the other important aspects of nutrition in other places that it is not a significant factor for me to take it out of my diet.”
		Yes	
30	Both	Somewhat	“Yeah, I would still say that they are the three most prominent, although they (the graphics) did add something to my knowledge of red meat in terms of health benefits, but I wouldn’t say that it would overall sway my current status. In terms of reduction, I was perhaps leaning more on the benefits or the health side of things vs. the cost would help me reduce to maybe one or two servings, but just trying to weigh those three factors.”
		Yes	
31	Both	Somewhat	“I don’t know because I find statistics hard to say show me a graphic and a number, I find it hard to relate to. I think maybe humans are bad at statistics and making decisions based on statistical data. If there were particular things related to my health like cholesterol levels of like actual diagnostics I can do on me to maybe say red meat isn’t helping, so it would have to be some kind of individual health need as opposed to some big external data graphic, I think.
		Yes	
32	Both	Somewhat	“Yeah, I was thinking about the amount of protein that I need to eat to maintain my health and the certainty level did not convince me to reduce. Like the factors are more important to me than reducing my cancer risk because I’m healthy right now and I feel like that would reduce some of my risk already, so I’m not overly concerned about cancer.”
		Yes	

^1^ Main determinant refers to whether participant valued the factors such as taste, cost, health, etc., or the evidence presented in MAGICapp graphics or a combination of the two when deciding their willingness to reduce their meat intake.

## Data Availability

Additional data is available upon.

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
