# Peer review of "Values and Preferences Related to Cancer Risk among Red and Processed Meat Eaters: A Pilot Cross-Sectional Study with Semi-Structured Interviews"

_foods, 2021, doi:10.3390/foods10092182_

Round 1
Reviewer 1 Report
The manuscript is interesting and well written. However, there are a few confused descriptions of data. Misdescriptions should be corrected as indicated.
1. (Page 7): It seems that light bars (to eliminate) and dark bars (to reduce) are interchanged in Figure 1.
Correct the explanation of Figure 1.
2. There are a few confused descriptions of data. Correct the following sentences as shown in Figure 1 or Figure 2.
(Page 7): Fifteen (48.4%) of 31 participants were willing to potentially reduce their red meat consumption. Six (19%) were "definitely willing" to reduce (6 or 7 on the Likert scale), 9 (29%) participants were "somewhat willing” to reduce eating red meat (4 or 5 on the Likert scale), while the remaining 17 (46%) were “somewhat unwilling” to “definitely unwilling” to reduce their red meat consumption (Figure 1).
(Page 8): Eight (53.3%) of 15 participants were potentially willing to reduce their processed meat consumption. Four (26.7%) of 15 participants were “definitely willing” to reduce, while the remaining four (26.7%) participants were “somewhat willing” to reduce eating processed meat (Figure 2).
3. You mentioned that 31 of the 32 participants consumed at least 1 weekly serving of red meat. If so, what does ‘An equal number of these participants were male and female’ mean? Please clarify.
(Page 8): An equal number of these participants were male and female. Half were also between 18 and 40 years old, with one participant between 41 and 60 years old and the remaining five in the 61-80 year old category.
Reviewer 2 Report
This is a very interesting article on perceptions on cancer risk and red meaat consumption.
Even if the idea and the whole article are interesting, the development of the article must be improved.
- Introduction. Should be improved (maybe adding one or two more paragraphs on the social importance on meat consumption?)
- Methods. Good development. I can recommend a couple of references that can help to the qualitative-interviews part:
Hubert, A. Qualitative Research in the Anthropology of Food: A Comprehensive Qualitative/Quantitative Approach, in MacClancy, J. & Macbeth H. (eds) Researching Food habits. Methods and Problems. Oxford, Berghahn, 2004.
Medina, F. Xavier. ‘Tell me what you eat and you will tell me who you are’: Methodological Notes on the Interaction between Researcher and Informants in the Anthropology of Food, in MacClancy, J. & Macbeth H. (eds) Researching Food habits. Methods and Problems. Oxford, Berghahn, 2004.
Results: Even if very interesting, the results can be improved and developed.
Conclusions: MUST be developed. Too short, after a lot of interesting information.
It requires just a bit of more work.
Reviewer 3 Report
Please see the attachment
